# Engagement in HIV services over time among young women who sell sex in Zimbabwe

Sue Napierala[1]*, Sungai T. Chabata[2,3], Calum Davey[4], Elizabeth Fearon[5], Joanna Busza[6], Phillis Mushati[2], Owen Mugurungi[7], Karin Hatzold[8], Valentina Cambiano[9], Andrew Phillips[9], James R. Hargreaves[6], Frances M. Cowan[2,10]

1 Women's Global Health Imperative, RTI International, Berkeley, California, United States of America, 2 Centre for Sexual Health and HIV/AIDS Research (CeSHHAR) Zimbabwe, Harare, Zimbabwe, 3 Department of Public Health, Erasmus MC, University Medical Center Rotterdam, Rotterdam, The Netherlands, 4 Department of Public Health, Environments and Society, London School of Hygiene and Tropical Medicine, London, United Kingdom, 5 Department of Global Health and Development, London School of Hygiene and Tropical Medicine, London, United Kingdom, 6 Centre for Evaluation, London School of Hygiene and Tropical Medicine, London, United Kingdom, 7 Ministry of Health and Child Care, Harare, Zimbabwe, 8 Population Services International, Cape Town, South Africa, 9 Institute for Global Health, University College London, London, United Kingdom, 10 Department of International Public Health, Liverpool School of Tropical Medicine, Liverpool, United Kingdom

* snapierala@rti.org

**Data Availability Statement:** All relevant data are within the paper.

**Funding:** This research was supported by the UN Population Fund (Cowan - through Zimbabwe's Integrated Support Fund funded by UK Department

## Abstract

### Introduction

Young female sex workers (FSW) are disproportionately vulnerable to HIV. Zimbabwe data show higher HIV incidence and lower engagement in services compared to older FSW. Utilizing data from a combination HIV prevention and treatment intervention, we describe engagement in the HIV services over time among FSW 18–24 years, compared to those ≥25 years of age.

### Materials and methods

Data were collected via respondent-driven sampling (RDS) surveys in 14 communities in 2013 and 2016, with >2500 FSW per survey. They included blood samples for HIV and viral load testing. As the intervention had no significant impact on HIV care cascade outcomes, data were aggregated across study arms. Analyses used RDS-II estimation.

### Results

Mean age in 2013 and 2016 was 31 and 33 years, with 27% and 17% aged 18–24 years. Overall HIV prevalence was 59% at each timepoint, and 35% and 36% among younger FSW. From 2013 to 2016 there was an increase in young HIV-positive FSW knowing their status (38% vs 60%, OR = 2.51, p<0.01). Outcomes for all FSW improved significantly over time at all steps of the cascade, and the relative change over time was similar among older versus younger FSW for most cascade variables.

### Discussion

Young FSW had improvements in care cascade outcomes, and proportionate improvements similar to older FSW, yet they remain less engaged in services overall. This implies

for International Development, Irish Aid, and Swedish International Development Cooperation Agency), and the National Institute of Mental Health (Napierala - 1K01MH110316). The funders had no role in study design, data collection and analysis, decision to publish, or preparation of the manuscript.

**Competing interests:** The authors have declared that no competing interests exist.

that the dedicated FSW services in Zimbabwe are having a comparably positive impact across age groups, however more is likely required to address young FSW's unique vulnerabilities and needs.

## Introduction

Young female sex workers (FSW) are a highly vulnerable population who are disproportionately affected by HIV. Adolescents and young women who sell sex represent a substantial proportion of the FSW population. Globally data indicate that 20–40% of FSW initiate sex work as adolescents <20 years [1], while data from Zimbabwe estimate that over 40% begin selling sex at <25 years of age [2]. Young FSW face structural barriers to positive sexual health outcomes including legal issues, stigma, discrimination and violence, as well as behavioral and biological barriers related to both youth and their engagement in sex work. This results in increased susceptibility to HIV infection and lower engagement in HIV services as compared to their older counterparts [3–5].

Data from Zimbabwe and other countries are limited but imply high HIV incidence among younger FSW. Programmatic data between 2009–2014 from Zimbabwe's National Sex Worker Programme, *Sisters with a Voice* (*Sisters*), estimate HIV incidence of over 10% annually among FSW ≤25 years of age, as compared to 6% for those 36 years and older [6]. This significant HIV burden continues to influence the dynamic of country epidemics. Studies have also demonstrated lower engagement in HIV services by younger FSW as compared to those who are older, including being less likely to use condoms, to engage in PrEP, to know their HIV status, to be on antiretroviral therapy (ART), and to be virally suppressed [2,5,7,8].

Data on how best to offer HIV and other sexual health services to young FSW is emerging. Guidance on how to optimize young FSW engagement in HIV services has been outlined by UNAIDS and others [5,7–11]. Strategies include addressing structural barriers alongside delivery of behavioral and biomedical interventions, involving young FSW in program design and implementation, and taking into account specific contexts in which young women work. In addition, strategies including peer led and age matched peer interventions, accessible confidential comprehensive clinic services, community support, and addressing structural drivers as well as sexual health needs have shown promise. However, there are few examples of successful programs targeting this population and rarely have programs been taken to scale.

We conducted the Sisters Antiretroviral therapy Programme for Prevention of HIV–an Integrated Response (SAPPH-IRe) trial (Pan African Trials Registry [PACTR201312000722390]) among FSW in Zimbabwe [12]. SAPPH-IRe was a cluster randomized trial to determine the effectiveness of an enhanced community-based intervention to increase uptake, retention in care and adherence to antiretroviral-based prevention and treatment among FSW. The primary outcome was the proportion of all FSW who had a viral load ≥1000 copies/mL. We previously presented the HIV care cascade, comparing older versus younger FSW in the 2013 survey [2]. Here we utilize data from the 2013 and 2016 surveys of the SAPPH-IRe trial to describe the engagement over time of young FSW 18–24 years of age in HIV services and the HIV care cascade, and compare this to engagement by FSW ≥25 years.

## Materials and methods

The *Sisters with a Voice* (*Sisters*) programme offers health services for women who self-identify as FSW based on guidance from the World Health Organization (WHO) [13]. This includes

dedicated FSW clinics across the country offering HIV testing and counselling, reproductive health services, condom provision and health education supported by trained peer educators, and a program of participatory activities to build community empowerment.

Between November and December 2013, FSW were recruited to the baseline survey of the SAPPH-IRe trial using respondent-driven sampling (RDS) from 14 *Sisters* sites in Zimbabwe. Between April and May 2016 FSW were recruited to the endline survey from the same *Sisters* sites using identical techniques. RDS is a recommended sampling strategy for hard-to-reach populations [14]. Methods are described in detail elsewhere, [12,15] and our results are reported using the STROBE-RDS reporting guidelines [16]. Participants at each survey were eligible if they were ≥18 years of age and currently working as a FSW, defined as having exchanged sex for money, goods or services in the past 30 days, and had lived at the site for at least the previous 6 months. At the time of each survey, we conducted geographic and social mapping at each of the 14 trial sites to identify 6–8 'seed' participants. Seeds were purposefully selected to represent all sub-populations within sex worker communities at each site. Each seed participant was interviewed and given two recruitment coupons to pass on to FSW in their social network. When women receiving a coupon attended the survey, they were then given two coupons to pass on to other FSW they knew, who worked in that location and who had not previously been recruited to the survey. Each participant was given US$5 remuneration for participating in the interview, and an additional US$2 for each peer recruited (up to a maximum of two). In all 14 sites a maximum of six iterations, or 'waves', of this process were performed, including initial seeds. Approximately 200 FSW were recruited into the study per site, per survey.

Participants completed an interviewer-administered questionnaire to collect data on demographics, sexual behaviour, sex work, HIV testing history and serostatus, uptake of HIV services, and ART use. Participants also provided a finger-prick dried blood spot sample for HIV testing, and if positive, for viral load (VL) measurement using the same sample.

## Measures

Young FSW were defined as 18–24 years of age (WHO upper age range for defining young people), and older FSW were defined as 25 years and older. Ever attending *Sisters* services was self-reported, and measured in the past 6 months at baseline and 12 months at endline surveys. Relationship with other FSW was also self-reported and was measured as good/very good versus neutral/bad/very bad/no relationship. Mental health was assessed using the validated Patient Health Questionnaire (PHQ-9), a set of nine questions about depression in the previous two weeks [17]. A score of nine or more out of the 20-item assessment was indicative of depressive disorder. ART use was self-reported. Knowing one's HIV status was defined as the combined outcome of reporting having previously received an HIV positive test result, or having received an HIV-negative test result in the 6 months prior to the survey. Finger prick blood samples were collected as dried blood spot (DBS) and used for HIV antibody testing (AniLabsytems EIA kit; AniLabsystems Ltd, OyToilette 3, FIN-01720, Finland). If HIV antibodies were detected then the sample was tested for quantitative HIV viral load using NucliSENS EasyQ HIV-1 v2.0. Viral suppression was defined as <1000 copies/mL.

## Statistical analysis

The SAPPH-IRe intervention had no significant impact on the primary outcome of the proportion of all FSW who had a viral load ≥1000 copies/mL, or on other HIV care cascade outcomes, overall or amongst young FSW, but it had an impact on engagement with services [18]. We therefore aggregated data across study arms, and adjusted our regression analyses for

study arm. The representativeness of our RDS surveys and further RDS diagnostics have been reported elsewhere [19]. Data were pooled from across the 14 survey sites. As recommended, seeds were dropped from all analyses. To replicate what the RDS-II analysis package performs we applied the svy command in Stata, probability weighting participants by the inverse of their network size, i.e. the number of women that each individual could have recruited, but adapted it for use with data pooled across sites. Because the distribution of network size differed across sites, we normalized the inverse network size by dividing it by the sum of inverse network sizes at each site. A fixed effect term for study site was included in analyses. Analyses were conducted using Stata 14.2.

Participant characteristics at each survey were analyzed descriptively, and stratified by younger versus older age. Differences in distribution of participant characteristics by age group at each time point were assessed using RDS-II weighted Chi-squared tests. Behavioral and biological prevention and care variables were likewise stratified by age group and analyzed descriptively. Differences in these variables at each time point were measured separately for young FSW and older FSW using RDS-II weighted logistic regression models with behavioral and biological prevention and care variables as outcome variables, and survey year as exposure, adjusting for trial arm with a fixed term for site [19]. We also compared the change over time among older FSW versus the change over time among younger FSW using the interaction between survey year and age group.

## Ethics

This research was reviewed and approved by the Medical Research Council Zimbabwe, Research Council of Zimbabwe, University College London, the London School of Hygiene and Tropical Medicine, and RTI International prior to initiating research activities. All participants provided written informed consent for study participation.

## Results

A total of 2722 participants were recruited into the surveys in 2013 and 2883 participants in 2016. After dropping 90 and 93 seeds in each survey, respectively, and an additional 15 and 49 participants missing key variables, we were left with 2617 analyzable participants in 2013 and 2791 analyzable participants in 2016. Mean age was 31 years (range 18–65) in 2013 and 33 years (range 18–75) in 2016. In 2013, 24% (n = 641) of the survey population was 18–24 years of age whereas in 2016 they comprised 17% (n = 494). Overall, HIV prevalence was 59% at both timepoints and was 35% and 36% among young FSW in 2013 and 2016, respectively. Participant characteristics at each survey stratified by age group are presented in Table 1. Within each age group there were significant differences in some variables between those participating in the survey in 2013 and 2016. These included an increase in number of years in sex work, a decrease in those reporting no religion, a decrease in the amount charged per client, an increase in alcohol consumption, and an increase in symptoms of depressive disorder. Client recruitment among younger FSW increased from bars/nightclubs/entertainment venues and decreased from marketplaces and streets, while older FSW decreased recruitment from bars/nightclubs/entertainment venues and increased from other venues. Additionally, amongst older FSW only, those who participated in the 2016 survey were more likely to have children, report differences in the proportion of earnings from sex work, and had a greater number of clients in the past week as compared to those in the 2013 survey.

Between 2013 and 2016, young FSW demonstrated significant change in several important outcomes related to the care cascade (Table 2). There was a significant increase in the proportion of women who had ever attended a Sisters clinic (15% vs 51%, OR = 11.5, 95%CI: 9.45–

**Table 1. Characteristics of female sex workers, by age and survey year.**

| Characteristic | 2013 survey | | 2016 survey | | Comparison p-value of 2013 vs 2016 among those 18–24 years | Comparison p-value of 2013 vs 2016 among those ≥25 years |
|---|---|---|---|---|---|---|
| | 18–24 years (N = 641) n (RDS-weighted %) | ≥25 years (N = 1976) n (RDS-weighted %) | 18–24 years (N = 494) n (RDS-weighted %) | ≥25 years (N = 2297) n (RDS-weighted %) | | |
| Age at start of sex work | | | | | 0.713 | 0.336 |
| <18 years | 159 (24.4) | 139 (6.3) | 127 (27.3) | 147 (7.0) | | |
| 18–24 years | 482 (75.6) | 670 (30.1) | 366 (72.7) | 787 (31.6) | | |
| 25–29 years | n/a | 614 (34.9) | n/a | 670 (29.0) | | |
| 30–39 years | n/a | 479 (25.2) | n/a | 586 (26.5) | | |
| ≥40 years | n/a | 74 (3.5) | n/a | 106 (5.9) | | |
| Number of years in sex work | | | | | 0.040 | <0.001 |
| 0–1 years | 189 (36.3) | 151 (9.8) | 112 (25.1) | 106 (5.7) | | |
| 2–4 years | 368 (52.1) | 587 (31.1) | 296 (58.6) | 667 (29.3) | | |
| 5–8 years | 77 (10.5) | 538 (25.2) | 78 (13.6) | 605 (24.6) | | |
| ≥9 years | 7 (1.1) | 700 (34.0) | 7 (2.7) | 918 (40.4) | | |
| Marital status | | | | | 0.444 | 0.109 |
| Divorced/separated | 378 (59.9) | 1,259 (62.5) | 309 (65.1) | 1484 (63.1) | | |
| Widowed | 19 (2.4) | 461 (25.6) | 9 (1.0) | 539 (25.5) | | |
| Never been married | 241 (37.3) | 238 (10.8) | 173 (33.2) | 238 (10.0) | | |
| Married/living together as if married | 3 (0.4) | 18 (1.0) | 3 (0.6) | 35 (1.4) | | |
| Number of children | | | | | 0.789 | 0.035 |
| 0 | 191 (32.8) | 294 (17.1) | 139 (30.7) | 312 (13.5) | | |
| 1 | 374 (57.6) | 1,043 (51.6) | 292 (56.0) | 1157 (49.9) | | |
| ≥2 | 76 (9.6) | 639 (31.4) | 63 (13.3) | 828 (36.6) | | |
| Highest level of education | | | | | 0.562 | 0.835 |
| No formal schooling | 9 (1.8) | 87 (5.1) | 6 (1.9) | 100 (6.1) | | |
| Some primary school | 140 (23.5) | 596 (35.5) | 99 (23.3) | 670 (33.2) | | |
| Some secondary school | 275(45.4) | 653 (31.5) | 203 (41.1) | 773 (33.7) | | |
| Completed secondary or more | 215 (29.3) | 625 (27.9) | 186 (33.7) | 753 (27.0) | | |
| Religion | | | | | <0.001 | <0.001 |
| Christian | 318 (50.8) | 1,161 (58.7) | 289 (54.9) | 1436 (60.7) | | |
| Other | 47 (8.3) | 223 (11.6) | 66 (14.4) | 336 (15.3) | | |
| No religion | 275 (40.9) | 590 (29.7) | 139 (30.7) | 525 (24.0) | | |
| Proportion of income generated through sex work | | | | | 0.117 | <0.001 |
| <25% | 35 (6.1) | 177 (9.5) | 22 (5.3) | 184 (10.1) | | |
| 25%-50% | 56 (10.2) | 243 (14.0) | 64 (13.9) | 373 (15.6) | | |
| >50%-99% | 157 (25.6) | 516 (25.3) | 123 (23.9) | 742 (31.4) | | |
| 100% | 393 (58.1) | 1,040 (51.2) | 285 (56.9) | 998 (42.9) | | |
| Venue for client recruitment | | | | | 0.007 | <0.001 |
| Bars/nightclubs/ entertainment venue | 471 (71.3) | 1,366 (68.6) | 336 (75.6) | 1351 (62.3) | | |
| By telephone | 35 (7.3) | 121 (6.3) | 17 (5.3) | 116 (5.8) | | |
| In the market place/ street | 98 (16.2) | 347 (19.5) | 66 (11.9) | 421 (18.6) | | |

(*Continued*)

**Table 1.** (Continued)

| Characteristic | 2013 survey | | 2016 survey | | Comparison p-value of 2013 vs 2016 among those 18–24 years | Comparison p-value of 2013 vs 2016 among those ≥25 years |
|---|---|---|---|---|---|---|
| | 18–24 years (N = 641) n (RDS-weighted %) | ≥25 years (N = 1976) n (RDS-weighted %) | 18–24 years (N = 494) n (RDS-weighted %) | ≥25 years (N = 2297) n (RDS-weighted %) | | |
| Other | 26 (5.3) | 96 (5.6) | 41 (7.2) | 272 (13.3) | | |
| Number of clients in the last week | | | | | 0.220 | 0.021 |
| 0 | 36 (6.1) | 160 (9.6) | 28 (7.3) | 139 (6.2) | | |
| 1–4 | 228 (41.6) | 798 (43.8) | 170 (33.9) | 941 (45.2) | | |
| 5–9 | 161 (25.2) | 481 (24.2) | 150 (28.4) | 619 (25.7) | | |
| ≥10 | 216 (27.1) | 537 (22.4) | 146 (30.4) | 598 (22.9) | | |
| Amount charged per client | | | | | 0.003 | <0.001 |
| ≤$2 | 5 (1.4) | 16 (1.1) | 1 (0.3) | 28 (1.2) | | |
| $2–5 | 380 (56.4) | 1,141 (61.4) | 335 (69.9) | 1742 (77.3) | | |
| >$5–10 | 207 (34.2) | 693 (33.6) | 142 (27.3) | 465 (20.0) | | |
| >$10 | 38 (8.1) | 78 (3.9) | 12 (2.5) | 37 (1.5) | | |
| Alcohol consumption in the past 12 months | | | | | 0.001 | <0.001 |
| Never | 222 (38.2) | 780 (43.7) | 151 (32.3) | 908 (40.4) | | |
| Once a month or less | 63 (10.5) | 200 (11.1) | 37 (6.6) | 228 (10.3) | | |
| 2–4 times per month | 88 (13.0) | 304 (15.2) | 54 (10.8) | 299 (13.4) | | |
| 2–3 times per week | 100 (14.7) | 238 (10.7) | 126 (27.2) | 393 (16.1) | | |
| 4 or more times per week | 168 (23.6) | 451 (19.4) | 126 (23.1) | 467 (19.8) | | |
| Symptoms of depressive disorder† | | | | | <0.001 | <0.001 |
| Yes | 253 (41.2) | 878 (45.8) | 252 (55.1) | 1413 (62.0) | | |
| No | 387 (58.8) | 1084 (54.2) | 240 (44.9) | 881 (38.0) | | |

†As per the Patient Health Questionniare (PHQ-9).

14.65, p<0.001), however this was measured in the last 6 months in 2013 versus 12 months in 2016. We saw significantly more young FSW who knew their HIV status overall (either positive or negative; 70% vs 79%, OR = 1.69, 95%CI: 1.10–2.59, p = 0.016), as well as more women living with HIV who knew their HIV-positive status (38% vs 60%, OR = 2.51, 95%CI: 1.40–4.47, p = 0.002). Among young FSW living with HIV, we also saw significant increases in other cascade outcomes of ART use (21% vs 41%, OR = 2.90, 95%CI: 1.54–5.49, p = 0.001) and viral suppression (13% vs 31%, OR = 3.74, 95%CI: 1.70–8.22, p = 0.001) among those reporting being on ART. The proportional increases of these two variables among young FSW, 20% and 18% respectively, were similar to the increases seen among older FSW of 23% for both outcomes. Among all young FSW living with HIV, the proportion with viral load <1000 copies regardless of reporting ART use also increased significantly from 31% in 2013 to 48% in 2016 (OR = 2.13, 95%CI: 1.12–4.05, p = 0.021). Older FSW also demonstrated significant improvements and similar proportional increases in all cascade variables over time. Importantly, when comparing the change over time among older versus younger FSW, we only found a significant difference in the proportion who had attended a Sisters clinic (OR = 1.66, 95%CI: 1.03–2.69, p = 0.037) among the variables evaluated.

In Fig 1, we compared changes in the care cascade in the 2013 and 2016 surveys by age group. Proportionately there are greater increases at all steps of the care cascade among young

**Table 2. Comparison of changes over time among older FSW and younger FSW.**

| | 2013 survey (N = 2617) | | | | 2016 survey (N = 2791) | | | | 2016 vs 2013 among 18–24 year olds | | 2016 vs 2013 among ≥25 year olds | | Change among older vs change among younger FSW | |
| | 18–24 years (N = 641) | | ≥25 years (N = 1976) | | 18–24 years (N = 494) | | ≥25 years (N = 2297) | | | | | | | |
| | n/N | RDS-weighted % (95% CI) | n/N | RDS-weighted % (95% CI) | n/N | RDS-weighted % (95% CI) | n/N | RDS-weighted % (95% CI) | OR* (95% CI) | P-value | OR‡ (95% CI) | P-value | OR (95% CI) | P-value |
|---|---|---|---|---|---|---|---|---|---|---|---|---|---|---|
| **Attended Sisters clinic‡** | 112/641 | 15.4 (11.9–19.8) | 433/1976 | 19.3 (16.9–22.0) | 276/494 | 51.4 (45.1–57.6) | 1652/2297 | 67.3 (64.4–70.11) | 6.18 (4.10–9.30) | <0.001 | 11.5 (9.35–14.65) | <0.001 | 1.66 (1.03–2.69) | 0.037 |
| **Mean number of visits to the Sisters clinic‡§** | 112 | 2.8 (2.4–3.2) | 433 | 3.2 (2.9–3.5) | 276 | 10.0 (8.5–11.4) | 1652 | 9.1 (8.5–9.6) | 7.16 (4.89–9.42) | <0.001 | 5.84 (4.78–6.90) | <0.001 | 3.36 (0.30–37.61) | 0.323 |
| **Proportion know status** | 382/531 | 70.1 (64.1–75.6) | 1339/1792 | 73.3 (70.2–76.2) | 354/439 | 79.1 (73.6–83.8) | 1846/2098 | 87.8 (85.6–89.7) | 1.69 (1.10–2.59) | 0.016 | 2.65 (2.07–3.38) | <0.001 | 1.55 (0.97–2.46) | 0.065 |
| **Proportion report positive of positives** | 97/229 | 37.9 (29.3–47.3) | 913/1311 | 68.5 (64.6–72.1) | 84/169 | 59.8 (49.5–69.3) | 1221/1466 | 81.0 (77.7–83.9) | 2.51 (1.40–4.47) | 0.002 | 1.97 (1.51–2.57) | <0.001 | 0.76 (0.41–1.42) | 0.396 |
| **Proportion report on ART of positives** | 49/229 | 20.5 (13.8–29.3) | 625/1311 | 47.2 (43.2–51.2) | 62/169 | 40.9 (30.9–51.9) | 1060/1466 | 69.7 (66.1–73.1) | 2.90 (1.54–5.49) | 0.001 | 2.63 (2.08–3.32) | <0.001 | 0.93 (0.46–1.88) | 0.840 |
| **Proportion report on ART with VL<1000 copies/mL of positives** | 23/229 | 12.7 (7.1–21.7) | 463/1311 | 36.7 (32.9–40.8) | 42/169 | 30.8 (21.4–42.1) | 905/1466 | 60.4 (56.6–64.0) | 3.74 (1.70–8.22) | 0.001 | 2.67 (2.12–3.37) | <0.001 | 0.82 (0.35–1.92) | 0.655 |
| **Proportion of positives with VL<1000 copies/mL** | 68/229 | 31.2 (22.8–41.0) | 654/1311 | 51.1 (47.1–55.1) | 74/169 | 47.8 (37.3–58.5) | 1050/1466 | 71.9 (68.5–75.1) | 2.13 (1.12–4.05) | 0.021 | 2.46 (1.96–3.10) | <0.001 | 1.20 (0.62–2.31) | 0.585 |

†OR adjusted for study arm and site.

‡6 months reference period for 2013 and 12 months reference period for 2016.

§N mean (95% CI) & ttest p-values.

FSW as compared to older FSW. There were statistically similar improvements in engagement in both age groups at most steps of the care cascade. Despite this, younger FSW remain significantly less engaged at each step of the care cascade in 2016, as they were in the 2013 survey.

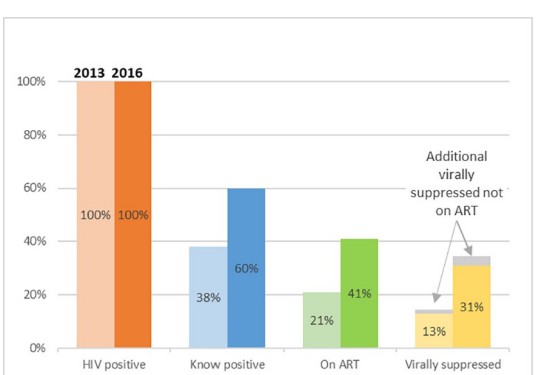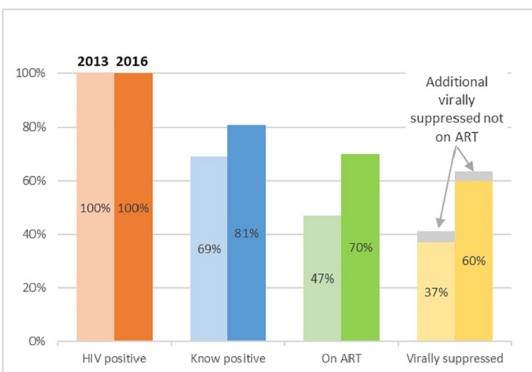

**Fig 1.** The HIV care cascades for younger (A) and older (B) FSW, in 2013 and 2016. (A) Women aged 18–24 (B) Women aged 25 and over.

## Discussion

Among this robust sample of over 2500 FSW from 14 sites nationally per survey, we found that by 2016 there were still just 37% of younger FSW living with HIV who were virally suppressed. It is critically important to improve this and other HIV outcomes in this population. However, we also found that this represents a significant improvement as compared with our 2013 findings. This is likely as a result both of general treatment scale up over this period, and more targeted efforts by the *Sisters* programme. A follow up look at cascade variable in this group will be very valuable.

We saw a large increase in ever attending a *Sisters* clinic among young FSW between 2013 and 2016 (15% and 51%), though we also inquired about a longer period of time (6 months in 2013 versus 12 months in 2016). Among older FSW this number increased more significantly from 19% to 67%. Literature from a number of settings indicate that both young people and FSW are reluctant to attend general public clinics [4,20]. Therefore, the relatively high overall level of clinic engagement, particularly among young FSW, is an encouraging sign which may indicate a broad knowledge and good reputation of the services provided by *Sisters* clinics in these communities. A more recent analysis of *Sisters* clinic attendance from 2009–2018 estimated that over 57% of all FSW working in Zimbabwe in 2017 had attended the programme, and the proportion of younger FSW has steadily increased over the period evaluated [21].

Our findings demonstrated encouraging improvements in a number of HIV care-related outcomes among young FSW, including improved engagement in each step of the HIV care cascade. Importantly, absolute change over time in these care cascade variables was similar among older and younger FSW, and was proportionally greater among young FSW. It is noteworthy that guidelines for ART eligibility changed between 2013 and 2016, which likely made more younger FSW eligible for ART based on CD4 count, contributing to increased engagement. Nonetheless, this implies that the dedicated services available to FSW in Zimbabwe are having a positive impact across age groups. However, younger FSW remain significantly less engaged in HIV care services as compared to their older counterparts. In terms of the global 90-90-90 goals, [22] per our data in 2016 younger FSW were at 60-68-76 and older FSW were at 81-86-86. Younger women remain underserved and more is likely required to address their unique vulnerabilities and needs.

Knowing one's HIV status is the critical first step to engagement in prevention and care services. The most pronounced drop-off in engagement in the HIV care cascade in both younger and older FSW was knowledge of HIV-positive status, with 60% and 81%, respectively, being aware of their positive status in 2016. This drop-off is common across settings in FSW populations as well as in the general population [2,23,24]. Efforts are underway to address this in many settings, through identifying differentiated testing options which will reach regular testers in a cost-effective manner, and providing a range of alternative outreach strategies to reach those reluctant to test regularly. Indeed, the *Sisters* programme has implemented a number of strategies to facilitate testing and improve knowledge of HIV status, including outreach testing services, self-testing, appointment reminders, peer education programming and non-stigmatizing services. This undoubtedly has contributed to the significant gains in overall knowledge of HIV status, defined as knowing one's positive status or having tested negative in the past 6 months. However, with these efforts we see differentiated improvement over time by age group. Young women have a 9% gain in knowledge of HIV status, while we see a 15% gain among older women (p = 0.065). Importantly though, we see a greater increase in the proportion of young FSW living with HIV who know their status than among older FSW, 22% (absolute increase) versus 13%, respectively, which is an encouraging finding. Notably, it is easier for a person living with HIV to know their status as it does not require recent testing, so we would expect higher values of those knowing their positive status in a group with higher

prevalence, i.e. the older FSW. However, in previously published research emerging from the baseline data we also identified a high reported testing frequency among young FSW, even higher than their older counterparts [2]. In the endline survey we had a similar finding.

An interesting finding were the changes over time in some behavioral characteristics, including increased alcohol consumption, a decrease in the amount charged per client, and an increase in symptoms of depressive disorder. While these changes were not differential by age group, it is worth considering what they might indicate about changes in the broader sex work environment, for example if they reflect underlying economic or social issues. It is also worth considering potential relationships between these factors and care cascade variables. Exploration of the implications of these changes were beyond the scope of this paper, but further research may be merited.

There were several limitations to this research. While RDS is an established technique for recruiting hidden populations, we cannot be certain of how representative our samples were at each timepoint and recruitment may have been subject to bias. We conducted analysis of potential biases which were largely reassuring though these analyses are not definitive [12]. We recruited up to six 'waves' of recruitment in each of the 14 RDS surveys at each timepoint. More waves imply deeper penetration into a population and thus the potential for more representative sampling. However, we identified differences in the characteristics of our study population across survey timepoints, and we recruited more young FSW in 2013 compared to 2016 (24% vs 17%). We cannot be certain whether this is due to fundamental changes in the population surveyed over time, or if this indicates that we recruited slightly different populations in the two surveys. For many of our outcomes, including knowledge of HIV status, we relied on self-reported data, which was thus subject to social desirability bias. However, we used rigorous biological measurement of HIV infection and viral load. We found a small proportion of women living with HIV who reported not being on ART yet were virally suppressed (see Fig 1). This suggests an under-reporting of ART use. This problem has been cited in several other studies using self-reported cascade indicators [25–29]. Social desirability bias and misunderstanding of survey questions or concepts are posited as possible explanations.

For both older and younger FSW, HIV programming should place special emphasis on engagement in HIV testing, and establish strong support systems for linkage to care. A critical element in reaching younger FSW will also be to identify strategies to reach women during the early years of their engagement in sex work, when they may be least likely to access targeted programs. Given the impressive gains along the HIV care continuum, the *Sisters* programme is a strong model for how to do this. For younger FSW, though we saw encouraging improvements in our outcomes of interest they remain less engaged overall than their older counterparts. This implies that additional initiatives may be required to reach and engage this population, for whom routine testing is particularly important and prevention services are critical. The *Sisters* program has now trained younger peer educators, is providing educational and other benefits to young women who sell sex through DREAMS[30,31] and other programs[32,33] and is trialing a microplanning programme, as methods to bridge this gap and increase engagement with HIV services overall. These and other innovative strategies are required to address the unique needs of younger FSW. Tailoring programs to address the needs of young FSW is critical to health equity and for reaching our global HIV goals.

## Acknowledgments

We thank the participants involved in this study and the community advisory boards for their time and contribution. We are also grateful to the data safety and monitoring board for their guidance and insight.

## Author Contributions

**Conceptualization:** Sue Napierala, Joanna Busza, Owen Mugurungi, Karin Hatzold, Valentina Cambiano, Andrew Phillips, James R. Hargreaves, Frances M. Cowan.

**Data curation:** Sungai T. Chabata, Calum Davey, Elizabeth Fearon, Valentina Cambiano, James R. Hargreaves, Frances M. Cowan.

**Formal analysis:** Sungai T. Chabata.

**Funding acquisition:** Frances M. Cowan.

**Investigation:** Sue Napierala, Andrew Phillips, James R. Hargreaves, Frances M. Cowan.

**Methodology:** Sue Napierala, Calum Davey, Elizabeth Fearon, Joanna Busza, Karin Hatzold, Valentina Cambiano, Andrew Phillips, James R. Hargreaves, Frances M. Cowan.

**Project administration:** Phillis Mushati, Owen Mugurungi, Frances M. Cowan.

**Resources:** Karin Hatzold.

**Supervision:** Phillis Mushati, Karin Hatzold, Frances M. Cowan.

**Validation:** Elizabeth Fearon, Andrew Phillips, Frances M. Cowan.

**Writing – original draft:** Sue Napierala.

**Writing – review & editing:** Sue Napierala, Sungai T. Chabata, Calum Davey, Elizabeth Fearon, Joanna Busza, Phillis Mushati, Karin Hatzold, Valentina Cambiano, Andrew Phillips, James R. Hargreaves, Frances M. Cowan.

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
