## [Decision Letter · Decision Letter 0]

5 Apr 2022

PONE-D-22-00643Engagement in HIV services over time among young women who sell sex in ZimbabwePLOS ONE

Dear Dr. Sue Napierala

Thank you for submitting your manuscript to PLOS ONE. After careful consideration, we feel that it has merit but does not fully meet PLOS ONE’s publication criteria as it currently stands. Therefore, we invite you to submit a revised version of the manuscript that addresses the points raised during the review process.

We look forward to receiving your revised manuscript.

Kind regards,

Carlos Miguel Rios-González, Ph.D

Academic Editor

PLOS ONE

Journal Requirements:

3. . In your Data Availability statement, you have not specified where the minimal data set underlying the results described in your manuscript can be found. PLOS defines a study's minimal data set as the underlying data used to reach the conclusions drawn in the manuscript and any additional data required to replicate the reported study findings in their entirety. All PLOS journals require that the minimal data set be made fully available. For more information about our data policy, please see http://journals.plos.org/plosone/s/data-availability.

Reviewers' comments:

Reviewer's Responses to Questions

**Comments to the Author**

1. Is the manuscript technically sound, and do the data support the conclusions?

Reviewer #1: Yes

Reviewer #2: Yes

Reviewer #3: Yes

2. Has the statistical analysis been performed appropriately and rigorously? 

Reviewer #1: Yes

Reviewer #2: Yes

Reviewer #3: Yes

3. Have the authors made all data underlying the findings in their manuscript fully available?

Reviewer #1: Yes

Reviewer #2: No

Reviewer #3: Yes

4. Is the manuscript presented in an intelligible fashion and written in standard English?

Reviewer #1: Yes

Reviewer #2: Yes

Reviewer #3: Yes

5. Review Comments to the Author

Reviewer #1: This paper follows up from a previously published manuscript reporting on the 2013 baseline survey of the SAPPH-IRe trial, which identified differences between young and older FSW across all pillars of the HIV care cascade. The current study explores how uptake of services and treatment outcomes has changed both within and between young and older FSW from 2013 to 2016. The findings demonstrate that even with encouraging improvements in both population groups, there are still concerning gaps in HIV service uptake and outcomes among younger FSW.

The findings add to growing recognition and reporting of differences between younger and older key population groups, but the value may be somewhat diminished by the publication delay. In the six years since the endline year (2016), changes in standard care practices have been introduced in Zimbabwe and beyond, such as universal test and treat, new approaches to testing such as HIV self testing, and differentiated service delivery for ART. These include strategies designed to better engage young people, as also outlined by the authors. Nonetheless, the findings suggest that even with continued improvements across the board in testing and treatment uptake and outcomes, engagement may remain lower among young FSW than older FSW. Differential and focused approaches to meet the needs of young FSW remain paramount.

The authors acknowledge a key limitation of the study, that recall periods for attending a Sisters service were different in baseline and endline surveys. However, the difference in proportion who have ever attended in past 6 months in 2013 (15%) and past 12 months in 2016 (51%) is substantial enough that one expects there would still be a significant increase if the recall period were identical.

Overall the paper is very well written and easy to follow.

Minor comments

Line 67 – believe it should read “are emerging”.

The outcome measure is described as “ever attending Sister services” (e.g. line 114, 175), yet neither survey actually measured lifetime attendance. Would it be more appropriate to phrase it as “recently attended a Sisters clinic”, maintaining the qualifier that it was measured in the last 6 months in 2013 versus 12 months in 2016?

Reviewer #2: This manuscript evaluated Engagement in HIV services over time among young women who sell sex in Zimbabwe. Authors provided reasonable background information of the subject matters and justified the relevance of the study.

Comment:

Abstract

Define RDS on its first appearance

Main article

Page 7, line 122 dried blood spot (D\\BS), I am not sure you wanted the back slash

Page 7, line 125, Better to say viral load suppression but not undetectable. The lower limit detection of viral load for most machines is set at 50, 40, 20. At 1000, you still detect viruses

Table1:

I will be better to rea arrange the columns such that

for 2013 arrange as 18-24 vs. 25 or older followed by its corresponding P-value. Do the same for 2016

Page 15 line 192: A couple of areas need to change verbs to past tense eg we compare, we see etc. This is true in some other areas as well

Page 15: There is title for the figure, however the figure is at the end. You may cite in the text and let the figure at the end, however, you also have tables in the text. Perhaps you should also move the figure up.

Figure: Need to improve its quality

Reviewer #3: ce manuscrit un peu ancien apporte des informations importantes pour la réponse au sida au Zimbabwe. En fait il montre bien que les services effectués sur les populations des jeunes travailleuses de sexe ont un impact et méritent d'être poursuivis.

Des compléments importants méritent d'être apportés sur ce travail. Notamment des précisions sur les questions éthiques. Peu de choses sont dites sur les règles éthiques appliquées. prière de donner les références de la clearance éthique.

Pour le reste , la méthodologie utilisée est conforme aux populations difficiles d'atteindre et les résultats sont bien discutés.

6. PLOS authors have the option to publish the peer review history of their article (what does this mean?). If published, this will include your full peer review and any attached files.

Reviewer #1: No

Reviewer #2: No

Reviewer #3: **Yes: **Serge Clotaire BILLONG

---

## [Author Response · Author response to Decision Letter 0]

2 May 2022

Response to reviewers' comments:

We thank the reviewer for their comments and suggestions. We have included a point-by-point response to their review in bold below.

This manuscript evaluated Engagement in HIV services over time among young women who sell sex in Zimbabwe. Authors provided reasonable background information of the subject matters and justified the relevance of the study.

Abstract

Define RDS on its first appearance

As suggested, we have added (RDS) after the instance where we mention respondent-driven sampling in the abstract, under Materials and Methods on line 34.

Main article

Page 7, line 122 dried blood spot (D\\BS), I am not sure you wanted the back slash

We appreciate the reviewer catching this typo. We have deleted the “\\” where indicated.

Page 7, line 125, Better to say viral load suppression but not undetectable. The lower limit detection of viral load for most machines is set at 50, 40, 20. At 1000, you still detect viruses

We thank the reviewer for this comment. We have eliminated mention of undetectable viral load, and changed the sentence in question to read: “Viral suppression was defined as <1000 copies/mL.”

Table1:

I will be better to rea arrange the columns such that

for 2013 arrange as 18-24 vs. 25 or older followed by its corresponding P-value. Do the same for 2016

For consistency with the other Tables and Figures in this manuscript, as well as with the text itself, we have opted to retain the use of ≥25 years, rather than changing Table 1 to read ’25 and older’.

Page 15 line 192: A couple of areas need to change verbs to past tense eg we compare, we see etc. This is true in some other areas as well

We have changed the tense of verbs in the present to past tense, per reviewer comment.

Page 15: There is title for the figure, however the figure is at the end. You may cite in the text and let the figure at the end, however, you also have tables in the text. Perhaps you should also move the figure up.

The placement of figures and tables are per Author Guidelines for this journal, therefore we have not made any change to address this comment. We are happy to submit in a different format if recommended by the editor.

Figure: Need to improve its quality

We have re-attached our figure with this re-submission. We believe the quality of our figure is per the standards of this journal, but please let us know if quality is an issue.

---

## [Editor Report · Decision Letter 1]

8 Jun 2022

Engagement in HIV services over time among young women who sell sex in Zimbabwe

PONE-D-22-00643R1

Dear Dr. Sue Napierala,

We’re pleased to inform you that your manuscript has been judged scientifically suitable for publication and will be formally accepted for publication once it meets all outstanding technical requirements.

Kind regards,

Carlos Miguel Rios-González, Ph.D

Academic Editor

PLOS ONE

---

## [Editor Report · Acceptance letter]

20 Jun 2022

PONE-D-22-00643R1 

Engagement in HIV services over time among young women who sell sex in Zimbabwe 

Dear Dr. Napierala:

I'm pleased to inform you that your manuscript has been deemed suitable for publication in PLOS ONE. Congratulations! Your manuscript is now with our production department. 

Kind regards, 

on behalf of

Dr. Carlos Miguel Rios-González 

Academic Editor

PLOS ONE